# Plant Milk-Clotting Enzymes for Cheesemaking

**DOI:** 10.3390/foods11060871

**Published:** 2022-03-18

**Authors:** Fabrizio Domenico Nicosia, Ivana Puglisi, Alessandra Pino, Cinzia Caggia, Cinzia Lucia Randazzo

**Affiliations:** 1Department of Agricultural, Food and Environment, University of Catania, 95123 Catania, Italy; fabrizio.nicosia@phd.unict.it (F.D.N.); ipuglisi@unict.it (I.P.); alessandra.pino@unict.it (A.P.); cinzia.caggia@unict.it (C.C.); 2ProBioEtna, Spin-off of University of Catania, 95123 Catania, Italy

**Keywords:** milk-clotting, plant proteases, proteolytic activity, cheese, enzymes

## Abstract

The reduced availability and the increasing prices of calf rennet, coupled to the growing global demand of cheese has led, worldwide, to explore alternative clotting enzymes, capable to replace traditional rennet, during the cheesemaking. In addition, religious factors and others related to the vegetarianism of some consumers, have led to alternative rennet substitutes. Nowadays, several plant-derived milk-clotting enzymes are available for cheesemaking technology. Many efforts have also been made to compare their effects on rheological and sensory properties of cheese to those arising from animal rennet. However, vegetable clotting enzymes are still partially suitable for cheesemaking, due to excessive proteolytic activity, which contribute to bitter flavor development. This review provides a literature overview of the most used vegetable clotting enzymes in cheese technology, classified according to their protease class. Finally, clotting and proteolytic activities are discussed in relation to their application on the different cheesemaking products.

## 1. Introduction

Cheesemaking is a dynamic process in which different technological steps, such as heat treatment, homogenization, and milk coagulation can affect the structure of the milk and determine the characteristics of the final product. Milk coagulation is a crucial step in cheesemaking and the choice of the specific clotting enzyme is fundamental for cheese yield, texture, and flavor. The widely used milk-clotting enzyme is chymosin (EC 3.4.23.4), an aspartic protease, which can hydrolyze a specific peptide bond (Phe_105_-Met_106_) present in κ-casein [1]. The C-terminal region of κ-casein is typically hydrophilic and has a negative charge at the native pH of milk (pH 6.7). It covers the casein micelles forming a layer that provides the stability of the micelles through steric hindrance and electrostatic repulsion, once hydrolyzed, the first phase of renneting begins [2]. Casein micelle core is formed by submicells of α- and β-caseins linked together by hydrophobic interactions and by insoluble calcium phosphate, which represents a cross-linking agent [3].

During cheesemaking, the reduction in milk pH by starter cultures leads to the solubilization of a part of the calcium phosphate, which is released as Ca^2+^. The latter directly influences the second phase of renneting, further reducing the colloidal stability of casein micelles and leading the aggregation of the micelles in the forming curd [4,5,6].

To predict the clot formation and the characteristic flavors and body-texture of the final product, the milk-clotting activity (MCA) and proteolytic activity (PA) ratio should be evaluated. MCA refers to the specificity of hydrolysis of the clotting enzyme towards κ-casein, whereas PA refers to the hydrolysis of proteins present in the curd (mainly consisting of α_sl_-, α_s2_-, β-, and κ-casein), which can lead to the formation of aftertaste in the long term. A ratio similar to chymosin (that is considered as a reference) corresponds to a high-quality coagulant, reflecting a high curd yield and low cheese defects, such as bitter flavors and an excellent final product with desirable firmness [7]. Figure 1 illustrates the mechanism of the milk-clotting enzyme process that generates bitter flavor.

Traditionally, chymosin extracted from the abomasum of calves has been used for cheesemaking. However, the reduced availability of calf rennet coupled to the increasing price and growing demand for cheese, have led to search for rennet substitutes coagulants. In fact, nowadays, calf rennet only covers 20–30% of the world demand for milk coagulants [8]. In addition, the market is increasingly directed towards diversification of supply to ensure consumers a wide choice, so cheese industries search for alternative coagulants to satisfy consumers who for religious (Islam, Judaism) and ethical reasons (vegetarians) prefer not to consume cheese made with animal rennet. Among the most commercially used substitutes for animal rennet, there are microbial coagulants: aspartic proteases produced by *Rhizomucor miehei* and *Rhizomucor pusillus*, which have a three-dimensional structure similar to chymosin, capable of hydrolyzing, in the same way, the Phe_105_-Met_106_ bond of κ-casein [9]. The aspartic proteases produced by these filamentous fungi are called mucorpepsin (EC 3.4.23.23) and are synthesized as a precursor: the protease obtained from *R. pusillus* consists of 437 amino acids, 22 of which are signal peptides; the following 44 amino acids are propeptide, and 361 amino acids constitute a mature protease. The precursor of the protease from *R. miehei* comprises a signal peptide of 22 amino acids, a propeptide comprising of 47 amino acids, and a mature protease made up of 361 amino acids [10]. Another aspartic protease that originates from a microorganism is the endothia pepsin (EC 3.4.23.22) produced by *Cryphonectria parastica*. The precursor consists of 419 amino acids, the signal peptide consists of 20 amino acids, the propeptide of 69 amino acids, and the mature enzyme includes 330 amino acids. Unlike the proteases produced by *Rhizomucor*, this enzyme hydrolyzes a different site of the κ-casein, the Ser_104_-Phe_105_ bond [9]. Microbial coagulants have several advantages: low cost of production, and conformity with kosher, halal, and vegetarian eating principles. Their main disadvantages are low specificity, high thermal stability, lower MCA/PA ratio than calf rennet, and more chances of bitterness in resultant cheese [11]. It is advisable that the proteases have low thermal stability, in this way, by regulating the post-cheesemaking temperatures, the residual enzyme survival is reduced, avoiding unwanted proteolysis and, therefore, the development of defects [12]. However, the cheeses obtained using endothia pepsin, thanks to its high thermolability, were evaluated as equivalent or even superior in quality than control cheeses produced using animal chymosin [13]. One of the most innovative methods to solve the lack of animal rennet is represented by the Fermentation Produced Chymosin (FPC) or genetic chymosin, which is obtained from a host microorganism such as *Escherichia coli*, *Bacillus subtilis*, and *Lactococcus lactis*, in which the gene for the protease is expressed [14]. Cheese is the first food product made using the recombinant DNA technique recognized by the Food and Drug Administration (FDA) [14]. Through a reverse transcriptase process, the mRNA of chymosin from the animal abomasum is transferred to cDNA and subsequently grafted into the DNA of a Generally recognized as safe (GRAS) microorganism that will be able to produce chymosin through fermentation [8]. The products of fermentation contain chymosin identical to the animal source, meaning that they have the same amino acid sequence as chymosin from the corresponding animal stomach [15]. One of the major advantages of this process is that 100% pure chymosin can be obtained, whereas calf rennet is composed of about 80% chymosin and 20% pepsin (that is less specific in hydrolyzing caseins) [1]. One problem is the stringent regulations of some countries towards genetically engineered foods; in fact, FPC is banned in Germany, Netherlands, and France [16]. The milk-clotting enzyme market is mostly occupied by FPC and, according to statistics, the cheese produced through FPC in the United States and the United Kingdom comprises 70% and 90% of the total cheese production, respectively [17].

Alongside the microbial and recombinant milk-clotting enzymes, several animal source coagulants are available on the market and recently have been discussed by Liu and co-workers [17]. These animal coagulants are interesting rennet substitutes and are suitable for cheese production, generating functional peptides during cheese ripening and flavors compounds appreciated by consumers. However, greater clarity should be paid on cleavage sites and functional peptides after genetic modification [17].

## 2. Types and Characteristics of Vegetable Proteases

Recently great attention has increasingly shifted to coagulants of vegetable origin such as proteases present in various plant tissues such as cardoon flower [18], *Cynara scolymus*’ artichoke [19], and *Citrus aurantium* flowers [20]. These proteases are enzymes found in plant tissues and can hydrolyze milk caseins at different pH and temperature [21].

The use of vegetable coagulants in cheesemaking has very ancient origins. Homer already wrote in the Iliad that fig juice was able to curdle milk. Both Hippocrates in the 5th century BC and Aristotle in the 4th century BC wrote about the use of fig latex to coagulate milk, while Lucius Junius Moderatus Columella in his treatise on agriculture, De Re Rustica, in the 1st century BC, mentions for the first time the use as a coagulant of wild thistle flowers, of seeds of *Carthamus tinctorius* and thyme, claiming that the cheese obtained had an excellent flavor [22]. Starting from the second half of the 19th century, the evolution of rennet of animal origin led to a reduction in the use of vegetable coagulants; these, in fact, pass from a primary to a secondary role. Subsequently, interest in vegetable coagulants grew again, when there was a rapid increase in the consumption of cheese and a reduction in the availability of animal rennet [21].

Proteases are enzymes present uniformly in the tissues of plants; they take on various functions from germination to senescence processes [23]. Proteases are classified based on the amino acid residues involved in the catalytic site and are divided into cysteine, serine, aspartic, and metalloprotease (that possess a metal-ion cofactor in the catalytic site) [24]. Table 1 showcases the plant enzyme coagulants recently discovered with their strengths, weaknesses, and optimal pH/temperature parameters. The main proteases involved in dairy preparations belong to the first three groups and none from metalloprotease [21], and for some of them, the specific hydrolytic site is not yet available. Similar to chymosin, many vegetable proteases selectively hydrolyze the Phe_105_-Met_106_ κ-casein bond, while others hydrolyze different sites, such as protease extracted from *Solanum dubium* hydrolyses Ser_104_-Phe_105_ bond of bovine κ-casein while actinidin (protease from *Actinidia chinensis*), which probably hydrolyses the Arg_97_-His_98_ or Lys_111_-Lys_112_ bond [25,26]. Moreover, the extract from the ginger rhizome (*Zingiber officinale*) can hydrolyze κ-casein at two different sites: Ala_90_-Glu_91_ and His_102_-Leu_103_, instead of proteases from *Cynanchum otophyllum Schneid* hydrolyses specifically the Ser_132_-Thr_133_ bond [27,28]. The hydrolysis of κ-casein, which is essential to initiate the coagulation process during cheesemaking, directly affects the milk-clotting activity (MCA) of the enzyme, which is not influenced by the difference in the cleavage point [29]. On the other hand, the difference in hydrolysis specificity of the different proteases towards α- and β-casein contribute to the release of peptides that can influence the flavor and texture of the cheese [30]. For example, cardosin A (from *C. cardunculus*) acts on different bonds of α_s1_-casein: Phe_24_-Phe_25_, Arg_100_-Leu_101_, Phe_153_-Tyr_154_, Trp_164_-Tyr_165_, and Tyr_165_-Tyr_166_, also on β-casein: Leu_127_-Thr_128_, Leu_165_-Ser_166_, and Leu_192_-Tyr_193_ [31]. The protease from cock’s eggs (*Salpichroa origanifolia*) hydrolyses the Phe_23_-Phe_24_ and Trp_164_-Tyr_165_ bonds in α-casein and the Leu_192_-Tyr_193_ bond in β-casein [32]. These different sites of action of the proteases lead to the formation of peptides that can have functional aspects, such as antihypertensive, immunomodulating, and antithrombotic activity [22]. However, the general hydrolysis of caseins is related to the proteolytic activity (PA) of the enzyme. High PA may be associated with the development of bitter taste and cheese texture defects [15]. These properties and characteristics mainly depend on the activity of starter/adjunct culture, but also the type of enzyme used for cheesemaking influence them at the beginning of cheese ripening [33].

### 2.1. Aspartic Proteases

Aspartic proteases have two aspartic acid residues in the catalytic site, they involve a water molecule that acts as a nucleophile in the hydrolysis reaction [34]. Most aspartic proteases consist of single-chain enzymes with a molecular weight of about 35 kDa and a length of about 330 amino acids [35,36]. These types of proteases are most active at acid pH (pH value 3–5) [37] and consist of two lobes containing the aspartic acid residues mentioned above, which are essential for carrying out the catalytic function of the enzyme [38]. Aspartic proteases have a structure that consists almost entirely of β-sheet and minimally of α-helix [39]. These proteases have been found in many vegetable tissues, such as: *Citrus aurantium* flowers [20], *Withania coagulans* fruit [40], and *Cirisium vulgare* flowers [41]. The most widespread and used aspartic proteases are certainly the cardosins extracted from *C. cardunculus* [42]. The most abundant are cardosin A and cardosin B; although, more recently, four additional cardosins have been isolated from flowers (cardosins E, F, G, and H) [42,43]. Cardosin A and cardosin B are heterodimeric glycoproteins composed of a heavy (31 and 34 kDa) and a light (15 and 14 kDa) chain. Cardosin A and B are often compared to chymosin and pepsin, respectively, as cardosin B is less selective than A in hydrolyzing caseins [22]. However, a comparison between chymosin and cardosins shows that the latter have a greater PA, which leads to the onset of bitter flavors [44,45]. In the Iberian Peninsula, the *C. cardunculus* extract containing these proteases is used to produce many cheeses that have a characteristic soft creamy texture and delicate flavor, sometimes slightly bitter but piquant when more mature [46]. The industrial implementation of these proteases is inhibited by several factors such as the variability of these plant extracts in respect to enzymatic activity and the limited resources of flowers [45]. To overcome these limitations, the use of recombinant cardosins in *Escherichia coli* and in *Kluyveromyces lactis* has been studied [22]. Tito et al. [47] discovered two innovative aspartic proteases in *Solanum tuberosum* capable of hydrolyzing milk caseins; specifically, the preferred substrate is represented by β-casein followed by α- and κ-casein. Both enzymes exhibited MCA in a dose-dependent manner with an optimum value determined at pH 5 and 30 °C. Furthermore, the different hydrolysis sites cleaved by *Solanum tuberosum* with respect to chymosin indicate the possible generation of peptides that could contribute to new tastes and aromas in cheese. In a similar way the aspartic protease contained in the fruits of *Salpichroa origanifolia*, (it has a maximum of MCA at 40 °C, pH 6.0) hydrolyzing mainly the α-casein and forming peptides that can have many potential biological activities [32].

### 2.2. Cysteine Proteases

Cysteine-type proteases have Cys and His residues in their catalytic site and comprises of a total of 108 different families [48]. The active catalytic domain contains three catalytic residues (Cys-His-Asn) that have specific functions: the Cys residue acts as a nucleophile while the His residue acts as a general base for proton shuttling [49]. These types of enzymes are produced as inactive precursors and have a peptide that acts as a signal for the secretion of the protein and an auto-inhibitory prodomain to prevent unwanted protein degradation [50]. One of the most studied cysteine proteases is papain from *Carica papaya* [51], but other cysteine proteases have been extracted from ginger rhizomes [52], from the root latex of *Jacaratia corumbensis* [53], and from *Actinidia chinensis* [54]. Albuquerque de Farias et al. [55] found a cysteine protease present in the fruits of *Morinda citrifolia*. The enzyme extract of these fruits has an optimum temperature of 50 °C and a pH of 6.0. Cheeses produced with this extract exhibited significantly higher fresh weight and yield than cheeses produced with commercial calf rennet with similar chewiness, but lower hardness, cohesiveness, elasticity, adhesiveness, and gumminess. A cysteine protease (MW 25.8 kDa) with MCA was isolated from *Dregea sinensis* stems with good activity and a wide pH range. However, it showed an optimum temperature of 80 °C, which is not used in dairy preparations [56].

Finally, actinidin (EC. 3.4.22.14), is a cysteine protease abundant in kiwifruit, composed of 220 amino acid residues with a molecular mass of 23.5 kDa. This enzyme exhibited promising characteristics as a milk-clotting agent in cheese technology [54]. There are many advantages derived from the use of actinidin as a coagulant enzyme in the cheesemaking: high MCA/PA ratio; the ability of specific hydrolyzing; and developing less off-flavor notes, attributed to bitter peptides [57]. In addition, the preparation of an aqueous extract from kiwifruit is easier, faster, and cheaper than other plant coagulants [26]. Mazorra-Manzano et al. [58] compared proteases extracted from ginger (*Zingiber officinale*) named zingibaine, melon (*Cucumis melo*) named cucumisin, and kiwifruit (*Actinidia deliciosa*) named actinidin, revealing that actinidin has the highest MCA/PA ratio compared to other vegetable coagulants. Moreover, Puglisi et al. [54] showed that actinidin has an excellent MCA/PA ratio in production of mozzarella with kiwi juice as a coagulant without the onset of bitter flavors. The milk coagulation with actinidin created cheese with characteristics similar to that prepared with calf rennet. In fact, elasticity, cohesion, chewiness, and hardness of cheeses prepared using calf rennet and kiwi proteolytic enzyme extract were reported to be quite similar to each other. On the contrary, curds obtained using melon extracts had different textural properties as reflected by the low values of hardness, cohesiveness, chewiness, and springiness, possibly due to the higher PA and the lower MCA/PA ratio [58]. Similar results were obtained from a recent study conducted by Fguiri et al. [59], where the coagulating properties of kiwi extract were compared to those of ginger and pineapple extracts in camel (*Camelus dromedarius*) milk cheesemaking. The data reported that kiwi extract determined the highest yield (20.71%). Furthermore, the cheese made with the kiwi extract showed a better texture and had the highest scores in the test of sensory evaluation compared with other extracts.

### 2.3. Serine Proteases

Serine proteases are enzymes that involve a Ser residue in their catalytic site and are grouped into more than 20 families. Serine proteases form covalent enzyme/substrate complexes, they have strong nucleophilic amino acid residues in their catalytic site so perform a nucleophilic attack on the carbonyl group of the peptide bond of the substrate [60]. They are present in many plant tissues, but more abundantly in fruits, and take part in many metabolic pathways of the plant [61]. A serine protease capable of clotting milk was extracted from *Solanum dubium*, named dubiumin, which was reported to be very stable against a wide range of pH values (4.0–11.0) as well as a wide range of temperature (20–90 °C) [62]. Other plant serine proteases capable of coagulating milk are cucumisin from *Cucumis melo* [63], religiosin from *Ficus religiosa* [64] and streblin from *Streblus Asper* [65]. An alternative to animal rennet can be represented by the serine protease extracted from fennel’s (*Foeniculum vulgare*) tissues, this enzyme remains active and stable in the pH ranging from 6 to 7.5 and temperatures ranging from 40 to 60 °C [66]. In many cases, raw extracts from plants containing different types of proteases are used to coagulate the milk, this is the case of the extract of *Vallesia glabra*, where the different types of proteases influence the proteolytic profile of the extract in a very wide pH range (2.5–12.0) [67]. The different tissues extract of crown flower (*Calotropis gigantea*) contains several proteases: results reported that latex exhibited high caseinolytic (86.45 U/mL) as well as MCA (450 U/mL) when compared to other parts (stem, flower, and leaf) [68]. *Bromelia pinguin* owns cysteine and serine proteases in the extract that are responsible for the major MCA in a broad temperature range with milk coagulation times comparable with commercial chymosin, but which also involve high PA; the MCA/PA ratio is 209 for chymosin and 1.29 for *B. pinguin,* respectively [69]. Finally, *Balanites aegyptiaca* extract owes its proteolytic characteristics both to the class of aspartic proteases and to the serine class, which gives it two optimum pH values (pH 5.0 and pH 8.0, respectively) [70].

**Table 1 foods-11-00871-t001:** Strengths, weaknesses, and optimum pH/temperature of plant proteases recently discovered.

Plant Source	Tissues	Type	Strengths	Weaknesses	Temperature (°C)	pH	Reference
*Calotropis gigantea*	Latex, stem, flower, and leaf	CP, SP	Latex has the highest MCA/PA ratio	High rate of proteolysis of crude enzyme	37	5.5	[68]
*Citrus aurantium*	Flower	AP	Raw enzyme extract is capable of coagulating milk in similar times to that required by animal rennet	NR	65–70	4.0	[20]
*Zingiber officinale*	Rhizomes	CP	The vegetable coagulant is easily extracted through few purification steps	Further studies are needed for industrial application	60	5.5	[52]
*Silybum marianum*	Flower	AP	Proteases extracted from *Silybum marianum* clot bovine, caprine, and ovine milks	NR	NA	NA	[71]
*Balanites aegyptiaca*	Fruit	AP, SP	MCA was found from the extract in the fruit pulp	Further studies on the organoleptic acceptability of cheeses produced are necessary	50	5.0, 8.0	[70]
*Cynara scolymus*	Flower	AP	Cheese yield is similar to that of animal rennet	Prolonged brining period (40 h) is necessary to avoid the development of bitter flavors in the cheese	40–60	4.0	[72,73]
*Foeniculum vulgare*	Stems	SP	The proteases are active at the temperature and pH parameters used for cheesemaking	The extraction process is complex	37	6.4	[66]
*Dregea sinensis*	Stems	CP	Purified cysteine protease shows a wide range of activity (pH and temperature)	The optimum temperature is about 80 °C, which is not adopted in the cheesemaking process	80	6.0–9.0	[56]
*Bromelia pinguin*	Fruit	CP, SP	The enzyme extract is able to coagulate milk in a relatively short period of time	High caseinolytic activity after a long incubation period	45	2.5, 7.5	[69]
*Morinda citrifolia*	Fruit	CP	With an MCA value of 238.80 ± 5.29 U/mL, *Morinda citrifolia* fruit extract proves to be a good candidate to replace calf rennet	Slightly bitter taste but good acceptability of cheeses	50	6.0, 7.0	[55]
*Vallesia glabra*	Leaf, fruits, and seed	AP, CP, SP	The extract obtained from the leaves shows a great activity (0.20 MCU/mL) while in the fruits and seeds it was 0.12 and 0.11 MCU/mL, respectively	Further studies are needed to better characterize the wide variety of proteases present in the raw extract	65–70	4.0	[67]
*Solanum tuberosum*	Tubers and leaves	AP	The two aspartic proteases are able to operate at optimal cheesemaking conditions (temperature 40–42 °C, pH values 6–6.2)	NR	30	5.0	[47]
*Salpichroa origanifolia*	Fruit	AP	The activity of the enzyme allows to enrich the cheese with bioactive peptides deriving from the hydrolysis of α-, β-, and κ-casein, which provide a health-promoting effect	NR	40	6.0	[32]
*Ficus johannis*	Latex	CP	The low tendency to autolysis, that is, autodigestion during storage at room temperature, suggesting a probable use in industrial cheesemaking	Enzyme loses 20% activity at high salt concentrations (1 M NaCl)	60	6.5	[74]
*Solanum elaeagnifolium*	Fruit	NA	This plant-derived protease is characterized by a good MCA/PA ratio	High concentrations of this coagulant can negatively affect the visco-elastic properties of the cheese	45	6.0	[75]
*Actinidia chinensis*	Fruit	CP	Kiwi extract exhibits high MCA/PA ratio compared to other plant coagulants	NR	40	5.5	[58]
*Cynara cardunculus*	Hairy root cultures	AP, SP, CP	*Cynara cardunculus* roots can produce proteases such as cardosin A with MCA, the phenotypic characteristic of high growth could lead to continuous supply from an application point of view	The low concentration of these proteases prevents industrial implementation	NA	NA	[76]

AP: Aspartic protease, CP: Cysteine protease, SP: Serine protease, NA: not available, NR: not revealed.

## 3. Proteolytic Activity and Bitter Taste by Plant Derived Coagulant in Cheese

The milk-clotting is essential to start the coagulation process; therefore, it is necessary to know how to evaluate the activity of each enzyme. To measure this parameter there are various methods such as the Berridge, the Soxhlet, and the International method, whose units of measurement are, respectively, Berridge units, Soxhlet units, and International milk-clotting unit (IMCU) [15]. As reported in Table 2, these methods differ from each other, and it is not easy to be able to compare the various units of measurement found in the literature [15].

The high PA characterizes most of the vegetable proteases, which, especially during beginning of cheeses ripening, continue to hydrolyze the caseins causing the appearance of short peptides, which can have a bitter taste [80]. Many theories have tried to explain the origin of the bitter taste: i.e., the presence of hydrophobic amino acids inside the chain [81]; the presence in percentage of hydrophobic amino acids in the peptide side chains [82]. Specifically, the bitter taste is reduced when the hydrophobic amino acid is in a terminal position or it is completely free [83]. Ney gives a value to the average hydrophobicity of a peptide, called the Q value;
$$Q=\frac{\sum \Delta f}{n}$$

Δ*f* represents the sum of the solubility data (i.e., the sum of the free energies of transfer of the amino acid side chains) divided by the number (*n*) of amino acid residues. When this value exceeds 1400, the peptide is to be considered bitter, while below 1300, the peptide is not bitter, but the bitterness of a peptide that has a Q value between 1300 and 1400 cannot be predicted [82]. Therefore, the presence of specific amino acids with high hydrophobicity values such as lysine, leucine, and proline in the side chains of the peptides leads to the onset of the bitter taste. In the caseins’ micelles, there are large quantities of hydrophobic amino acids such as proline, valine, and leucine [84]. As mentioned above, completely bond-free hydrophobic amino acids do not have a bitter taste, but the presence of proline-rich oligopeptides in the side chains confers bitter taste during cheese ripening [85]. For this reason, the Q value turns out to be an important parameter for the identification of these peptides that are generated by the extended proteolysis of clotting enzymes and other endogenous proteases even by starter proteases [86]. Bitter peptides isolated from cheese are reported in Table 3. Hydrophobic amino acids are readily observed from both the C- and N-terminus of the peptide; all of them originated from α- or β-casein, as the result of secondary proteolysis [84].

## 4. Effect of Plant Coagulants on Cheeses

Enzymatic coagulation of milk is a key step in the cheesemaking. The evaluation of enzyme activities of vegetable coagulants and their comparison with those of commercial rennet (chymosin) is an important first step in selecting a suitable rennet substitute. In this context, many efforts have been made to establish the influence of plant coagulants on rheological properties, sensory characteristics (texture, flavor, taste, and color) as well as yield of cheese. Most of the studies have been carried out on traditional cheeses type, mainly produced in the Mediterranean countries, in West Africa, and in Southern Europe, under traditional procedures in small dairies or farms [46]. In Africa, there is an example of unripened cheese produced with the addition of a vegetable protease, the Warankashi, which is coagulated with the extract of Sodom apple leaf (*Calotropis procera*) [89]. Some Spanish, Portuguese, French, and Italian varieties of cheeses were produced with aqueous extracts of dried wild thistle flowers, of various species of the genus *Cynara* [46]. This plant species grows spontaneously, especially in the south-west of the Mediterranean regions in arid and uncultivated soils. There are many Portuguese and Spanish cheeses that have the Protected Designation of Origin (PDO) and Protected Geographical Indication (PGI) certifications, available on the market that are produced using *Cynara cardunculus* dried flower extracts as a coagulant. These cheese types are: Serra and Serpa for Portugal; Los Pedroches, La Serena, Torta del Casar (from ewe’s milk); and, also, Flor de Guía (from a blend of ewe’s and goat’s milk) from Spain [46]. A typical example of cheese produced with *C. cardunculus* extract is Los Pedroches, which takes its name from the area in the province of Cordoba (Spain). It is a hard, uncooked fatty cheese produced with raw Merino ewe’s milk, coagulated with a vegetable coagulant, and ripened for about 2 months [46]. During the cheese ripening, changes in the microbiological and physico-chemical parameters take place, giving a slightly spicy taste and a soft texture to the cheese [90]. This is a clear example of how plant proteases hydrolytic activity over caseins have a significant effect on curd and cheese properties. However, long exposition to proteases can cause proteolytic degradation of the casein network (especially α- and β-casein), thus reducing approximately 0.3–0.7% of the curd yield [8]. Furthermore, the amount of protease used for cheesemaking is very important, because insufficient quantities lead to softer consistencies of the cheese while an excess of protease causes secondary proteolysis and, therefore, development of bitter flavors [17]. This happens because in the secondary proteolysis, the nonspecific action of plant proteases against caseins produces peptides, which can be easily hydrolyzed into low molecular weight peptides (about 1400 Da) or into free amino acids by proteases of starter/adjunct cultures. In particular, low molecular weight peptides can have hydrophobic groups at C-terminal end, which give them a high Q value and therefore bitterness [24,91].

Indonesia produces Dangke, a cheese manufactured using papain (*Carica papaya*) as coagulant enzyme. The flavor of the cheeses prepared with different concentrations of papain was evaluated, and results clearly showed how the cheese obtained with a lower quantity of coagulant has a better flavor: each cheese was prepared using 2.5 L of milk, in which different amounts of papain were inoculated (0.06, 0.10, 0.14, and 0.16 g). The sample inoculated with 0.10 g of papain had the best results to the physical tests such as flavor, color finish, body, and texture score, probably because lower quantities of coagulant are associated with a lower PA [92]. Hence, the different hydrolysis rate of caseins by proteases can affect the cheese texture. According to the International Dairy Federation (IDF), cheeses have been classified based on their firmness into: extremely hard, hard, semi-hard, semi-soft, and soft cheese [93]. Table 4 shows the cheese texture in relation to the plant proteases used.

Parameters such as temperature range and enzyme concentration are fundamental in cheese manufacture [108]. A percentage between 15% and 30% of the milk coagulant remains active in the curd and influences characteristics and flavors of cheese, even more than the indigenous bacterial microbiota. This is the case of *C. cardunculus* extract used for La Serena cheese giving soft consistency and a slightly bitter, sometimes spicy aftertaste [24,101]. For these reasons plant coagulants, such as those obtained from *C. cardunculus*, can suitably replace animal rennet in the production of soft cheeses such as Roquefort and Serra da Estrela [46]. Therefore, although *Cynara cardunculus* proteases are suitable for all types of milk, cheeses prepared with cow’s milk always develop more bitter flavors than those made with ewe’s milk [109]. In fact, bovine and caprine caseins are more hydrolyzed by *C. cardunculus* extract than ovine caseins [24]. This is the main reason why, as shown in Table 4, many cheeses made with plant coagulant use ewe’s milk as a substrate. However, the main issue remains in its high PA, which causes an intense bitter flavor, not always appreciated by consumers [110]. In a study of Martínez-Ruiz et al. [104], the berries of *Solanum elaeagnifolium* (trompillo or silverleaf nightshade) in Chihuahua, northern Mexico, have been used in artisanal cheese named Asadero. Cheeses were obtained by a standardized process (32 °C milk temperature, 40 min renneting time), just changing the enzyme source. This type of cheese is softer than those made with chymosin due to their higher water content and proteolysis, and shows a shelf-life of 28 days at 4–6 °C. In addition, Domiati cheese in the work of Darwish [96] was characterized by particular texture and flavor due to the action of vegetable proteases from *Abizia lebbeck* and *Helianthus annuus* seeds. The cheeses made with these coagulants were prepared in milk heated to 50 °C and salted using sodium chloride to give a final concentration of 12%. The results showed that hardness, adhesiveness, gumminess, and chewiness of the cheeses produced with the plant coagulant were lower than those of control cheeses produced with chymosin. An important result obtained by Hashim et al. [52] using ginger extract (*Zingiber officinale*) in the preparation of Peshawari cheese (semi-hard, fresh cheese made from whole or semi-skimmed buffalo or cow’s milk). The cheeses obtained with the plant extract did not tend to be bitter and exhibited more appreciable sensory characteristics than those obtained with calf rennet, making ginger extract a promising substitute of animal coagulant. A similar result was obtained by Bruno et al. [111] in the production of cheese with *Bromelia hieronymi* fruit extract that did not show bitter taste. Although the use of plant coagulants on an industrial scale is limited by some defects, a reorganization of the cheesemaking parameters could lead to the minimization of off-flavors to obtain a better quality of the final product [38].

## 5. Strategy to Improve the Use of Plant Derived Coagulant

One of the most important challenges of plant proteases is to overcome the problem of extract variability in composition, which involves a difficult standardization and limits the industrial implementation [22]. The variability on composition derives from many factors such as the different location and composition of proteases within plant tissues [112].

For example, cardosin A accumulates more in the upper part of the *C. cardunculus* flower, in the protein storage vacuoles of the stigmatic papillae, while cardosin B is distributed outside the cell wall and in the lower part of the pistil [113]. The concentration of cardosins increase with the senescence of the flower and the MCA of the extracts may depends on the plant ecotype used [113,114]. To reduce the variability on composition, the extraction/purification techniques such as “salting out” (i.e., the precipitation of proteins in solution by adding ammonium sulphate) coupled to chromatographic methods (for the subsequent purification of the desired enzyme) could be applied. This process allows to isolate the desired enzyme from the rest of the compounds present in the aqueous extract (sugars, vitamins, water, etc.) [40,53]. Lyophilization is another approach to standardize plant aqueous extract. Recently a Spanish patent focused on the development of a powder extract of *C. cardunculus*, revealing that the extract becomes more hygienic, manageable, and stable during storage [46]. Another solution is represented by the use of recombinant cardosin, being the best way to produce large quantities of these proteases at an industrial level, with best result for cardosin B [22]. However, the characteristically low MCA/PA ratio of plant proteases produces undesirable effects in cheese, which can be managed in various ways. A possible strategy to improve the industrial use of vegetable coagulants could be the use of adjunct bacterial cultures with specific aminopeptidases, able to hydrolyze the bitter peptides formed during cheese ripening. Some lactic acid bacteria (LAB) such as strains belonging to *Lactococcus lactis* species, possess a complex proteolytic system capable of hydrolyzing the peptides, resulting in a bitter taste. From this point of view, the aminopeptidase N (PeP N) is one of the most important peptidases capable to hydrolyze peptides with hydrophobic N-terminal amino acid (Leu, Ala) [115]. Other peptidases able to reduce bitter peptides formation responsible for bitter taste are: PepI, PepP, PepQ, PepR, and PepX. Each of these peptidases are able to hydrolyze the proline in a specific position within the peptide, reducing the bitter taste. These peptidases were already found in strains belonging to *Lactococcus lactis* and *Lactobacillus helveticus* [116].

Another strategy to use plant coagulant with reduced bitter compounds formation is to ultrafiltrate cow’s milk to keep a concentration in solids (fat, proteins, and minerals) similar to ewe’s milk [117]. In fact, the application of *Cynara cardunculus* proteases to ultrafiltered milk resulted in a semi-hard cheese with sensory properties similar to those of cheese made with animal rennet. Another method to reduce the defects caused by the excessive PA of the plant derived coagulating enzymes, could be represented by the use of both coagulating enzymes, with a high MCA/PA ratio, and techniques able to reduce the residual activity after milk-clotting. For example, one method is to apply high temperatures to the cheeses in order to inactivate the enzyme after milk-clotting. Unfortunately, this method is difficult to be applied because heat, besides denaturing the enzyme, halts the secondary proteolysis, and may cause a deterioration of cheese texture [118].

Furthermore, it is possible to apply HP (high pressure) to inactivate plant protease after milk coagulation. By applying a pressure of 600 Mpa combined with a moderate temperature of 40 °C for 35 min, cheeses without bitter flavors, unaltered color [118], and greater cohesiveness and hardness were obtained.

Finally, the modification of some cheesemaking and ripening parameters can allow avoiding the changes caused by plant coagulants. Lorente et al. [72] showed how the parameters related to brining time can affect the flavor of the Gouda-type cheese using the *Cynara scolymus* flower extract as milk coagulant. In detail, a curd salted in a brining NaCl solution of 22 °C for 40 h showed less bitter taste and organoleptic properties similar to cheeses manufactured with animal rennet. In fact, the high salt concentration allows the ionic strength to increase and, therefore, the hydrophobic interactions between the caseins to improve, thus inhibiting the hydrolysis of the hydrophobic regions of the caseins (such as C-terminal of β-casein) and releasing hydrophobic/bitter peptides (such as f193–209 from β-CN) [119].

## 6. Conclusions and Future Perspective

The present review summarized the main plant coagulants used in cheese technology and their effects on various cheese types. Based on the literature, most plant-derived clotting enzymes showed high PA, resulting in bitter compounds in the final products. For this reason, plant coagulants still have limitations in cheese production. Raw milk and technological cheese parameters should be carefully chosen when plant coagulants are used. For instance, based on literature, ewe’s milk is the most suitable for cheesemaking using plant derived proteases. Different sources of plant proteases (*Carica papaya*, *Ananas comosus*, *Ficus carica*, and *Cucumis melo*) with milk coagulating activity have been investigated. Unfortunately, these enzymes are characterized by high PA, and they can be used only for soft and creamy unripened cheese types. Among plant coagulants, actinidin is a promising rennet substitute for its specific hydrolytic site of action. Future researchers are required to better elucidate mechanisms involved in the casein hydrolysis and in peptide formation during the secondary proteolysis. The recent progress on recombinant form of some plant enzymes combined with more in-depth knowledge on the biochemical mechanism could help to expand the portfolio of rennet substitute coagulants in the dairy industry, to satisfy the increasing market segment of Kosher and Halal consumers.

## Figures and Tables

**Figure 1 foods-11-00871-f001:**
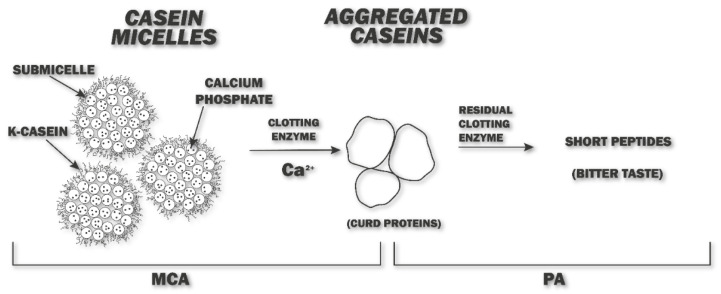
Degradation of caseins by clotting enzyme and development of bitter peptides. MCA = Milk-clotting activity, PA = Proteolytic activity.

**Table 2 foods-11-00871-t002:** Milk-clotting activity (MCA), proteolytic activity (PA), and MCA/PA ratio of plant extracts and other enzymes.

Plant Extract	MCA	PA	MCA/PA	Reference
*Calotropis gigantea*	450 (U/mL)	86.45 (U/mL)	5.21	[68]
*Zingiber officinale*	314 (unit/mg)	0.19 (unit/mg)	1653.00	[52]
*Silybum marianum*	0.083 (RU/mL)	0.128 (EA/mL)	0.65	[77]
*Balanites aegyptiaca*	2.43 (MCU/mL)	4.96 (MCU/mL)	0.49	[70]
*Cynara scolymus*	147.65 (MCU/mg)	5.45 (Ucas/mg)	27.1	[78]
*Bromelia pinguin*	2.59 (U/mg)	2.0 (U/mg)	1.29	[69]
*Morinda citrifolia*	238.8 (U/mL)	8.86 (U/mg)	27.00	[55]
*Vallesia glabra*	0.20 (U/mL)	19.04 (U/mL)	1.00	[67]
*Ficus johannis*	21.88 (U/mL)	0.339 (IU/mL)	64.54	[74]
*Solanum elaeagnifolium*	4347.00 (U/mL)	1.3 (U-Gly/mg)	3343.00	[75]
*Actinidia chinensis*	2.7 (U/mg)	0.55 (U/mg)	5.00	[58]
**Animal/microbial enzymes**				
Calf chymosin	551.00 (SU/mg)	2.28 (U/mg)	243.20	[79]
*R. miehei*	756.00 (SU/mg)	14.74 (U/mg)	51.31	[79]

**Table 3 foods-11-00871-t003:** Bitter peptides from Cheddar cheese.

Origin	Peptide	Q-Value	Reference
α_S1_-CN (f11–14)	Leu-Pro-Gln-Glu	1367	[84]
α_S1_-CN (f1–7)	Arg-Pro-Lys-His-Pro-Ile-Lys	1771	[84]
α_S1_-CN (f191–197)	Lys-Pro-Trp-Ile-Gln-Pro-Lys	2010	[84]
β-CN (f73–76)	Ile-Pro-Pro-Leu	2658	[87]
β-CN (f60–68)	Tyr-Pro-Phe-Pro-Gly-Pro-Ile-His-Asn	1871	[87]
β-CN (f8–16)	Val-Pro-Gly-Glu-Ile-Val-Glu-Ser-Leu	1390	[84]
β-CN (f200–206)	Val-Arg-Gly-Pro-Phe-Pro	1718	[87]
β-CN (f193–209)	Tyr-Gln-Glu-Pro-Val-Leu-Gly-Pro-Val-Arg-Gly-Pro-Phe-Pro-Ile-Ile-Val	1839	[88]

**Table 4 foods-11-00871-t004:** Cheese types in relation to plant coagulant.

Cheese Type	Name	Milk Type	Plant Coagulant Source	Reference
Soft	Torta del Casar	Ewe	*Cynara cardunculus*	[94]
	Dangke	Buffalo	*Carica papaya*	[95]
	Domiati	Buffalo	*Heliantus hannuus*	[96]
	Warankashi	Cow and Soymilk	*Calotropis procera* or *Carica papaya*	[89]
Semi-soft	Castelo Branco, Serra da Estrela, Serpa, Aizeitão, La Serena, Caciofiore dei Sibillini	Ewe	*Cynara cardunculus*	[97,98,99,100,101,102]
	Flor de Guía, Mestiço de Tolosa	Ewe and Goat	*Cynara cardunculus*	[22,103]
	Asadero	Cow	*Solanum elaeagnifolium*	[104]
Semi-hard	Los Pedroches, Évora Nisa,	Ewe	*Cynara cardunculus*	[105,106,107]

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
