# Peer review of "Plant Milk-Clotting Enzymes for Cheesemaking"

_foods, 2022, doi:10.3390/foods11060871_

Round 1

Reviewer 1 Report

Vegetable coagulants are promising milk-clotting enzyme for cheese-making technology

Suggested title: Delete ‘Technology’ at end of title

When recent 2 papers on same topic has been extensively reviewed, what was the need to choose the same topic once again:   Nitu et al. (2021), Ben Amira et al. (2017) and Shah et al. (2014); all three articles already documented by the authors

Suggested references to read and incorporate in Review paper

Folgado A and Abravles R. (2020) Plants, 9(2): 147 Industrial application of plant proteases (Thistle)

Mazorra-Manzano, MA et al. (2013) Food Chem. 141:1902-1907. Three plant protease extracts.

Grozdanovic MM et al. (2013) Int. Dairy J., 32(1): 46-52 Kiwifruit protease

Feijoo-Siofa & Villa (2011) Food and Bioprocess Technol., 4(6): 1066-1088 Genetically modified plant proteases.

Atigui, M. et al. (2021) J Chem., doi.org/10.1155/2021/6680246. Kiwi protease

Guevara and Daleo (2018). Book Ref. doi:10.1007/978-3-319-97132-2_2 (Biotechnology applications of plant proteolytic enzymes).

Suggestions

Try to utilize the abbreviations (i.e. MCA, PA) wherever used in text, once such abbreviation has been mentioned earlier in text

Suggest the usage rate for vegetable rennet as against animal (chymosin) rennet; especially since PA is greater than MCA in former rennets.

It would be better for the readers to know about MCA/PA ratio of Chymosin and even some important microbial rennets (i.e. derived from Mucor, Endothia).

Is there any test method to detect presence of vegetable (or even animal rennet – other than chymosin) to avoid the consumers getting cheated, when they claim vegetarian cheese (and use animal rennet instead)?

Any work of use of plant rennet on camel milk coagulation; camel milk is even difficult to be coagulated with chymosin.

Is there any commercial firm producing and marketing plant rennets anywhere in globe?

Figure 1: Include involvement of milk Ca2+ in secondary phase of rennet action to obtain the renneted cheese milk coagulum

Table 3: While printing fraction of specific casein subunit – mention ‘f’ before fraction of peptide number) i.e.  β-CN(f73-76)

Table 4: Club cheese types (for specific type – i.e. semi-hard/hard) when using same type of milk (i.e. Ewe milk) and same coagulant (i.e. Cynara cardunculus)

Conclusions

Write under this which milk (Ewe, cow, buffalo, camel, goat) will perform best in cheese making.

Which plant protease enzyme is most promising as rennet substitute in cheesemaking. Has such rennet been commercialized and industry using it anywhere in globe?

References

When writing reference for any Book – do mention the following: Publisher with place of publication, Name(s) of Editors, Edition number, Maybe Volume number, complete page range

Author Response

Vegetable coagulants are promising milk-clotting enzyme for cheese-making technology

Suggested title: Delete ‘Technology’ at end of title

When recent 2 papers on same topic has been extensively reviewed, what was the need to choose the same topic once again:   Nitu et al. (2021), Ben Amira et al. (2017) and Shah et al. (2014); all three articles already documented by the authors

Thanks for your comment. We modified the title as follow: “Plant milk clotting enzymes for cheesemaking”.

We agree with you about the existence of two recent review about similar topic. However, compared to both Shah et al. (2014) and Ben Amira et al. (2017), we reviewed more recent scientific paper (till 2021). In addition, compared to the review paper wrote by Nitu et al. (2021), we extensively argued about the technological application of different plant-based enzymes in terms of type of cheese; bitter taste formation and strategies to reduce bitter compounds.

Suggested references to read and incorporate in Review paper

Folgado A and Abravles R. (2020) Plants, 9(2): 147 Industrial application of plant proteases (Thistle)

Mazorra-Manzano, MA et al. (2013) Food Chem. 141:1902-1907. Three plant protease extracts.

Grozdanovic MM et al. (2013) Int. Dairy J., 32(1): 46-52 Kiwifruit protease

Feijoo-Siofa & Villa (2011) Food and Bioprocess Technol., 4(6): 1066-1088 Genetically modified plant proteases.

Atigui, M. et al. (2021) J Chem., doi.org/10.1155/2021/6680246. Kiwi protease

Guevara and Daleo (2018). Book Ref. doi:10.1007/978-3-319-97132-2_2 (Biotechnology applications of plant proteolytic enzymes).

Thanks for your suggestion. All suggested references were already cited in the review paper except for the work by Atigui, M. et al. (2021), wich was added in the text (lines 230-235).

Suggestions

Try to utilize the abbreviations (i.e. MCA, PA) wherever used in text, once such abbreviation has been mentioned earlier in text

Thanks for your comment. We used abbreviation thought the text.

Suggest the usage rate for vegetable rennet as against animal (chymosin) rennet; especially since PA is greater than MCA in former rennets.

Thanks for your suggestion. In many case the usage rate is not reported, except for papain in lines 334-341

It would be better for the readers to know about MCA/PA ratio of Chymosin and even some important microbial rennets (i.e. derived from Mucor, Endothia).

Thanks for your comment. We added information about MCA/PA ratio of both Chymosin microbial rennets in Table 2.

Is there any test method to detect presence of vegetable (or even animal rennet – other than chymosin) to avoid the consumers getting cheated, when they claim vegetarian cheese (and use animal rennet instead)?

We don’t know about the existance of a method able to define the nature of coagulant (vegetable or animal) used in the cheesemaking. Up to now, based on the EU regulation the type of coagulant, reported in the label, is indicated as animal or vegetable, including the microbial coagulant.  

Any work of use of plant rennet on camel milk coagulation; camel milk is even difficult to be coagulated with chymosin.

Thanks for your comment. We added the work of Atigui, M. et al. (2021) related to the use of plant rennet on camel milk coagulation (lines 230-235). Even camel milk is difficult to be coagulated by chymosin, the manuscript showed good promising results by using kiwi extract.

Is there any commercial firm producing and marketing plant rennets anywhere in globe?

Thanks for your comment. In Italy, Prodor srl, (Bobbio, Piacenza), produces a liquid extract of C. cardunculus and other plant enzymes called “Galium” (100g of this product is used to coagulate 100 liters of milk).

Figure 1: Include involvement of milk Ca2+ in secondary phase of rennet action to obtain the renneted cheese milk coagulum

Thanks for your comment. We included in Figure 1 involvement of milk Ca2+ in secondary phase of rennet action to obtain the renneted cheese milk coagulum.

Table 3: While printing fraction of specific casein subunit – mention ‘f’ before fraction of peptide number) i.e.  β-CN(f73-76)

Thanks for your comment. We mentioned ‘f’ before fraction of peptide number.

Table 4: Club cheese types (for specific type – i.e. semi-hard/hard) when using same type of milk (i.e. Ewe milk) and same coagulant (i.e. Cynara cardunculus)

Thanks for your comment. We clubbed cheese types when using same type of milk.

Conclusions

Write under this which milk (Ewe, cow, buffalo, camel, goat) will perform best in cheese making.

Thanks for your comment. We reported in the text that ewe’s milk is the most suitable for cheesemaking (lines 437-438).

Which plant protease enzyme is most promising as rennet substitute in cheesemaking. Has such rennet been commercialized and industry using it anywhere in globe?

Based on the review of the available literature, kiwi extract appears promising as rennet sustitute in cheesemaking. Up to now, based on our knowledge, no company commercializes the kiwi extract. We wish that both scineitifc community and food industry will focus the attention kiwi extract industrial implementation.   

References

When writing reference for any Book – do mention the following: Publisher with place of publication, Name(s) of Editors, Edition number, Maybe Volume number, complete page range

Thanks for your comment. We revised the reference style.

Reviewer 2 Report

In general, the article is interesting, but still requires some changes and supplements. Detailed comments can be found in the attached pdf file. 

Author Response

we modified the text according to your suggestion in the pdf attached file

Reviewer 3 Report

This manuscript reviewed various sources of plant coagulants for cheesemaking. Despite the subject is of interest, the manuscript should be improved in the order of sections to make it more attractive to readers. For instance, Since this paper intends to highlight potential of plant enzymes for cheese manufacture, First part of section 3 should be introductory to give readers an idea of important parameters regarding cheesemaking and proteolysis and their role on the final quality of a cheese product; then should describe main enzymes found (and make a comparison with animal or microbial origin and then describe various cheese varieties. Finally, authors should address major limitations found when used vegetable enzymes and then address strategies to solve problems (example bitterness), but based on what is generally seen in the literature (i.e., variability on composition, lack of standardization on cheesemaking, etc.). Cheese scientists would identify these problems first and then address cheese manufacture practices to standardize processing, besides proposing other alternatives (such as the listed in the manuscript).

Other comments can be found below:

English edition/revision is required.

Line 29: Should use symbol for k (kappa).

Line 13, 29: cheese-making processes does not have appropriate soundness right for English language and should be replaced.

Lines 62-64: Parragraph should contain reference, since there is no clarity if statements are personal opinions from authors, rather than a consensus from cheese/dairy scientists (e.g., genetic chymosin).

Table 1 could include information at what pH/temperatures these enzymes are active, so it can give an idea readers what type of cheese can potentially be made.

Author Response

This manuscript reviewed various sources of plant coagulants for cheesemaking. Despite the subject is of interest, the manuscript should be improved in the order of sections to make it more attractive to readers. For instance, Since this paper intends to highlight potential of plant enzymes for cheese manufacture, First part of section 3 should be introductory to give readers an idea of important parameters regarding cheesemaking and proteolysis and their role on the final quality of a cheese product; then should describe main enzymes found (and make a comparison with animal or microbial origin and then describe various cheese varieties. Finally, authors should address major limitations found when used vegetable enzymes and then address strategies to solve problems (example bitterness), but based on what is generally seen in the literature (i.e., variability on composition, lack of standardization on cheesemaking, etc.). Cheese scientists would identify these problems first and then address cheese manufacture practices to standardize processing, besides proposing other alternatives (such as the listed in the manuscript).

Thanks for your comment. As you suggested, we revided the order of sections. We introduced the important parameters regarding cheesemaking and proteolysis, giving to readers an idea of their role on the final product. In addition, we improved the text addressing major limitations found when vegetable enzymes are used,  focusing on strategies to solve problems.

Other comments can be found below:

English edition/revision is required.

Thanks for your suggestion. We revised the text by a native english.

Line 29: Should use symbol for k (kappa).

Thanks for your comment. We used the symbol for k.

Line 13, 29: cheese-making processes does not have appropriate soundness right for English language and should be replaced.

Thanks for your comment. We modified the text as you suggested.

Lines 62-64: Parragraph should contain reference, since there is no clarity if statements are personal opinions from authors, rather than a consensus from cheese/dairy scientists (e.g., genetic chymosin).

Thanks for your comment. We Clarified the statement reporting the reference.

Table 1 could include information at what pH/temperatures these enzymes are active, so it can give an idea readers what type of cheese can potentially be made.

Thanks for your comment. We included in Table 1 information about pH and temperature for all the reported enzymes.

Round 2

Reviewer 3 Report

Authors followed recooemndations and made a significant improvement. There are still few points they could address to make the manuscript more interesting:

Line 408: refer to cysteine-type instead?

Table 3: beta-casein f (193-209) is a common bitter peptide, chich is released by chymosin and should be considered in the list.

Ine 879: secondary proteolysis is usually associated with the nitrogen fraction comprised by low molecular size peptides and free amino acids usually released by action of microbial activity. Please clarify this actual difference with the occurrence of bitter peptides released by plant enzymes.

Line 1229-1234: authors should address how salt content in cheese can effectively reduce the incidence of bitter peptides when plant coagulants are used. This is a common approach observed when using chymosin as coagulant, where an increase of salt content can reduce activity of residual shymosin and therefore reducing the release of beta-casein f(193-209).

Author Response

Line 408: refer to cysteine-type instead?

Thanks for your comment. We modified the text as you suggested (line 190).

Table 3: beta-casein f (193-209) is a common bitter peptide, chich is released by chymosin and should be considered in the list.

Thanks for your suggestion. We added the peptide β-casein f (193-209) in table 3.

Ine 879: secondary proteolysis is usually associated with the nitrogen fraction comprised by low molecular size peptides and free amino acids usually released by action of microbial activity. Please clarify this actual difference with the occurrence of bitter peptides released by plant enzymes.

Thanks for your comment. We clarified this topic in lines 328-333.

Line 1229-1234: authors should address how salt content in cheese can effectively reduce the incidence of bitter peptides when plant coagulants are used. This is a common approach observed when using chymosin as coagulant, where an increase of salt content can reduce activity of residual shymosin and therefore reducing the release of beta-casein f(193-209).

Thanks for your suggestion. We explained how salt content can influence the developement of bitter peptides (lines 435-439).